# Recurrent Neural Language Models as Probabilistic Finite-state Automata

**Anej Svete**   **Ryan Cotterell**
{asvete, ryan.cotterell}@inf.ethz.ch
**ETH** zürich

## Abstract

Studying language models (LMs) in terms of well-understood formalisms allows us to precisely characterize their abilities and limitations. Previous work has investigated the representational capacity of recurrent neural network (RNN) LMs in terms of their capacity to recognize unweighted formal languages. However, LMs do not describe unweighted formal languages—rather, they define *probability distributions* over strings. In this work, we study what classes of such probability distributions RNN LMs can represent, which allows us to make more direct statements about their capabilities. We show that simple RNNs are equivalent to a subclass of probabilistic finite-state automata, and can thus model a strict subset of probability distributions expressible by finite-state models. Furthermore, we study the space complexity of representing finite-state LMs with RNNs. We show that, to represent an arbitrary deterministic finite-state LM with $N$ states over an alphabet $\Sigma$, an RNN requires $\Omega\left(N|\Sigma|\right)$ neurons. These results present a first step towards characterizing the classes of distributions RNN LMs can represent and thus help us understand their capabilities and limitations.

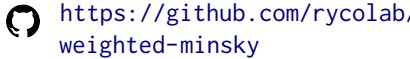
https://github.com/rycolab/
weighted-minsky

## 1 Introduction

We start with a few definitions. An **alphabet** $\Sigma$ is a finite, non-empty set. A **formal language** is a subset of $\Sigma$'s Kleene closure $\Sigma^*$, and a **language model** (LM) $p$ is a probability distribution over $\Sigma^*$. LMs have demonstrated utility in a variety of NLP tasks and have recently been proposed as a general model of computation for a wide variety of problems requiring (algorithmic) reasoning (Brown et al., 2020; Chen et al., 2021; Hoffmann et al., 2022; Chowdhery et al., 2022; Wei et al., 2022a,b; Kojima et al., 2023; Kim et al., 2023, *inter alia*). Our paper asks a simple question: How can we characterize the representational capacity of an LM based on a recurrent neural network (RNN)?

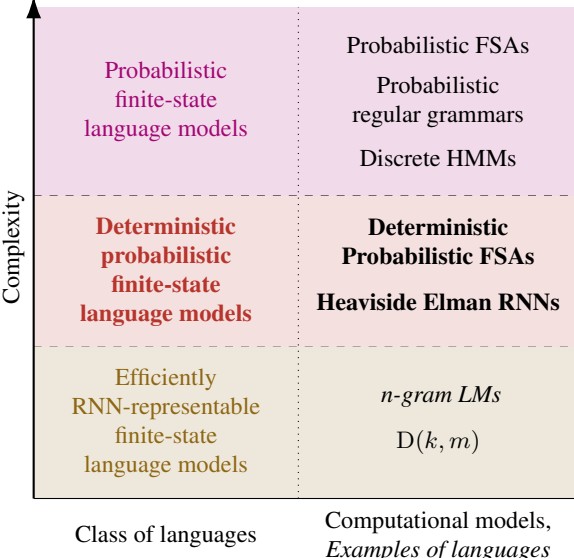

Figure 1: A graphical summary of the results. This paper establishes the equivalence between the **bolded** deterministic probabilistic FSAs and Heaviside Elman RNNs, which both define deterministic probabilistic finite-state languages.[1]

In other words: What classes of probability distributions over strings can RNNs represent?

Answering this question is essential whenever we require formal guarantees of the correctness of the outputs generated by an LM. For example, one might ask a language model to solve a mathematical problem based on a textual description (Shridhar et al., 2023) or ask it to find an optimal solution to an everyday optimization problem (Lin et al., 2021, Fig. 1). If such problems fall outside the representational capacity of the LM, we have no grounds to believe that the result provided by the model is correct in the general case. The question also follows a long line of work on the linguistic capabilities of LMs, as LMs must be able to implement mechanisms of recognizing specific syntactic structures to generate grammatical sequences (Linzen et al., 2016; Hewitt and Manning, 2019; Jawahar et al., 2019; Liu et al.,

---

[1] $\mathrm{D}(k, m)$ refers to the Dyck language of $k$ parenthesis types and nesting of up to depth $m$.

2019; Icard, 2020; Manning et al., 2020; Rogers et al., 2021; Belinkov, 2022, *inter alia*).

A natural way of quantifying the representational capacity of computational models is with the class of formal languages they can recognize (Deletang et al., 2023). Previous work has connected modern LM architectures such as RNNs (Elman, 1990; Hochreiter and Schmidhuber, 1997; Cho et al., 2014) and transformers (Vaswani et al., 2017) to formal models of computation such as finite-state automata, counter automata, and Turing machines (e.g., McCulloch and Pitts, 1943; Kleene, 1956; Siegelmann and Sontag, 1992; Hao et al., 2018; Korsky and Berwick, 2019; Merrill, 2019; Merrill et al., 2020; Hewitt et al., 2020; Merrill et al., 2022; Merrill and Tsilivis, 2022, *inter alia*). Through this, diverse formal properties of modern LM architectures have been shown, allowing us to draw conclusions on which phenomena of human language they can model and what types of algorithmic reasoning they can carry out.[2] However, most existing work has focused on the representational capacity of LMs in terms of classical, unweighted, formal languages, which arguably ignores an integral part of an LM: The *probabilities* assigned to strings. In contrast, in this work, we propose to study LMs by directly characterizing the class of probability distributions they can represent.

Concretely, we study the relationship between RNN LMs with the Heaviside activation function $H(x) \stackrel{\text{def}}{=} \mathbb{1}\{x > 0\}$ and finite-state LMs, the class of probability distributions that can be represented by weighted finite-state automata (WFSAs). Finite-state LMs form one of the simplest classes of probability distributions over strings (Icard, 2020) and include some well-known instances such as *n*-gram LMs. We first prove the equivalence in the representational capacity of deterministic WFSAs and RNN LMs with the Heaviside activation function, where determinism here refers to the determinism in transitioning between states conditioned on the input symbol. To show the equivalence, we generalize the well-known construction of an RNN encoding an unweighted FSA due to Minsky (1954) to the weighted case, which enables us to talk about string probabilities. We then consider the space complexity of simulating WFSAs using RNNs. Minsky's construction encodes an FSA with $N$ states in space $\mathcal{O}(|\Sigma|N)$, i.e., with an RNN with $\mathcal{O}(|\Sigma|N)$ hidden units, where $\Sigma$ is the alphabet over which

the WFSA is defined. Indyk (1995) showed that a general unweighted FSA with $N$ states can be simulated by an RNN with a hidden state of size $\mathcal{O}(|\Sigma|\sqrt{N})$. We show that this compression does not generalize to the weighted case: Simulating a weighted FSA with an RNN requires $\Omega(N)$ space due to the independence of the individual conditional probability distributions defined by the states of the WFSA. Lastly, we also study the asymptotic space complexity with respect to the size of the alphabet, $|\Sigma|$. We again find that it generally scales linearly with $|\Sigma|$. However, we also identify classes of WFSAs, including *n*-gram LMs, where the space complexity scales logarithmically with $|\Sigma|$. These results are schematically presented in Fig. 1.

## 2 Finite-state Language Models

Most modern LMs define $p(\boldsymbol{y})$ as a product of conditional probability distributions $p$:

$$p(\boldsymbol{y}) \stackrel{\text{def}}{=} p(\text{EOS} \mid \boldsymbol{y}) \prod_{t=1}^{|\boldsymbol{y}|} p(y_t \mid \boldsymbol{y}_{<t}), \quad (1)$$

where $\text{EOS} \notin \Sigma$ is a special end of sequence symbol. The EOS symbol enables us to define the probability of a string purely based on the conditional distributions.[3] Such models are called **locally normalized**. We denote $\overline{\Sigma} \stackrel{\text{def}}{=} \Sigma \cup \{\text{EOS}\}$. Throughout this paper, we will assume $p$ defines a valid probability distribution over $\Sigma^*$, i.e., that $p$ is tight (Du et al., 2023, §4).

**Definition 2.1** (Weakly Equivalent). *Two LMs $p$ and $q$ over $\Sigma^*$ are **weakly equivalent** if $p(\boldsymbol{y}) = q(\boldsymbol{y})$ for all $\boldsymbol{y} \in \Sigma^*$.*[4]

Finite-state automata are a tidy and well-understood formalism for describing languages.

**Definition 2.2.** *A **finite-state automaton (FSA)** is a 5-tuple $(\Sigma, Q, I, F, \delta)$ where $\Sigma$ is an alphabet, $Q$ a finite set of states, $I, F \subseteq Q$ the set of initial and final states, and $\delta \subseteq Q \times \Sigma \times Q$ set of transitions.*

We assume that states are identified by integers in $\mathbb{Z}_{|Q|} \stackrel{\text{def}}{=} \{0, \ldots, |Q|-1\}$.[5] We also adopt a more

---

[2]See §7 for a thorough discussion of relevant work.

[3]Sampling EOS ends the generation of a string, which makes EOS analogous to the *final weights* in a WFSA. We make the connection more concrete at the end of this section.

[4]We distinguish two notions of equivalence: *weak* and *strong* equivalence. The latter, informally, corresponds to the notion that there is a one-to-one correspondence between the sequences of actions performed by $p$ and $q$ to generate any string $\boldsymbol{y} \in \Sigma^*$. Naturally, strong equivalence implies weak equivalence.

[5]For a cleaner presentation, we also assume that vectors and matrices are zero-indexed.

suggestive notation for transitions by denoting $(q, y, q') \in \delta$ as $q \xrightarrow{y} q'$. We call transitions of the form $q \xrightarrow{y} q'$ $y$**-transitions** and define the **children** of the state $q$ as the set $\left\{ q' \mid \exists y \in \Sigma \colon q \xrightarrow{y} q' \in \delta \right\}$.

FSAs are often augmented with weights.

**Definition 2.3.** *A **real-weighted finite-state automaton** (WFSA) $\mathcal{A}$ is a 5-tuple $(\Sigma, Q, \delta, \lambda, \rho)$ where $\Sigma$ is an alphabet, $Q$ a finite set of states, $\delta \subseteq Q \times \Sigma \times \mathbb{R} \times Q$ a finite set of weighted transitions and $\lambda, \rho \colon Q \to \mathbb{R}$ the initial and final weighting functions.*

We denote $(q, y, w, q') \in \delta$ with $q \xrightarrow{y/w} q'$ and define $\tau(q \xrightarrow{y/w} q') \overset{\text{def}}{=} w$, where $\tau(q \xrightarrow{y/\circ} q') \overset{\text{def}}{=} 0$ if there are no $y$-transitions from $q$ to $q'$.[6] The **underlying FSA** of a WFSA is the FSA obtained by removing the transition weights and setting $I = \{q \in Q \mid \lambda(q) \neq 0\}$ and $F = \{q \in Q \mid \rho(q) \neq 0\}$.

**Definition 2.4.** *An FSA $\mathcal{A} = (\Sigma, Q, I, F, \delta)$ is **deterministic** if $|I| = 1$ and for every $(q, y) \in Q \times \Sigma$, there is at most one $q' \in Q$ such that $q \xrightarrow{y} q' \in \delta$. A WFSA is deterministic if its underlying FSA is deterministic.*

In contrast to unweighted FSAs, not all non-deterministic WFSAs admit a weakly equivalent deterministic one, i.e., they are non-determinizable.

**Definition 2.5.** *A **path** $\boldsymbol{\pi}$ is a sequence of consecutive transitions $q_1 \xrightarrow{y_1/w_1} q_2, \cdots, q_N \xrightarrow{y_N/w_N} q_{N+1}$. Its **length** $|\boldsymbol{\pi}|$ is the number of transition in it and its **scan** $\mathbf{s}(\boldsymbol{\pi})$ the concatenation of the symbols on them. We denote with $\Pi(\mathcal{A})$ the set of all paths in $\mathcal{A}$ and with $\Pi(\mathcal{A}, \boldsymbol{y})$ the set of all paths that scan $\boldsymbol{y} \in \Sigma^*$.*

The weights of the transitions along a path are multiplicatively combined to form the weight of the path. The weights of all the paths scanning the same string are combined additively to form the weights of that string.

**Definition 2.6.** *The **path weight** of $\boldsymbol{\pi} \in \Pi(\mathcal{A})$ is $\mathbf{w}(\boldsymbol{\pi}) = \lambda(q_1) \left[ \prod_{n=1}^{N} w_n \right] \rho(q_{N+1})$. The **string-sum** of $\boldsymbol{y} \in \Sigma^*$ is $\mathcal{A}(\boldsymbol{y}) \overset{\text{def}}{=} \sum_{\boldsymbol{\pi} \in \Pi(\mathcal{A}, \boldsymbol{y})} \mathbf{w}(\boldsymbol{\pi})$.*

A class of WFSAs important for defining LMs is probabilistic WFSAs.

**Definition 2.7.** *A WFSA $\mathcal{A} = (\Sigma, Q, \delta, \lambda, \rho)$ is **probabilistic (a PFSA)** if all transition, initial, and*

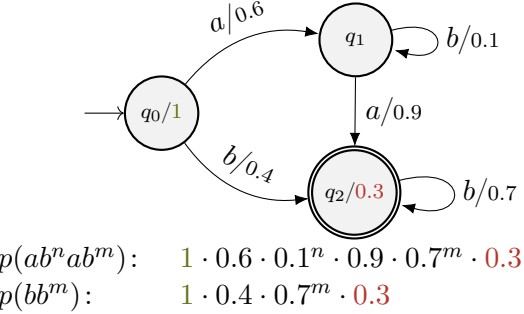

$$p(ab^nab^m)\colon \quad 1 \cdot 0.6 \cdot 0.1^n \cdot 0.9 \cdot 0.7^m \cdot 0.3$$
$$p(bb^m)\colon \quad 1 \cdot 0.4 \cdot 0.7^m \cdot 0.3$$

Figure 2: A weighted finite-state automaton defining a probability distribution over $\{a, b\}^*$.

*final weights are non-negative,* $\sum_{q \in Q} \lambda(q) = 1$, *and, for all $q \in Q$,* $\sum_{q \xrightarrow{y/w} q' \in \delta} w + \rho(q) = 1$.

The initial weights, and, for any $q \in Q$, the weights of its outgoing transitions and its final weight, form a probability distribution. The final weights in a PFSA play an analogous role to the EOS symbol—they represent the probability of ending a path in $q$: $\rho(q)$ corresponds to the probability of ending a string $\boldsymbol{y}$, $p(\text{EOS} \mid \boldsymbol{y})$, where $q$ is a state arrived at by $\mathcal{A}$ after reading $\boldsymbol{y}$. We will use the acronym DPFSA for the important special case of a deterministic PFSA.

**Definition 2.8.** *A language model $p$ is **finite-state** (an FSLM) if it can be represented by a PFSA, i.e., if there exists a PFSA $\mathcal{A}$ such that, for every $\boldsymbol{y} \in \Sigma^*$, $p(\boldsymbol{y}) = \mathcal{A}(\boldsymbol{y})$.*

See Fig. 2 for an example of a PFSA defining an FSLM over $\Sigma = \{a, b\}$. Its support consists of the strings $ab^nab^m$ and $bb^m$ for $n, m \in \mathbb{N}_{\geqslant 0}$.

In general, there can be infinitely many PFSAs that express a given FSLM. However, in the deterministic case, there is a unique minimal canonical DPFSA.

**Definition 2.9.** *Let $p$ be an FSLM. A PFSA $\mathcal{A}$ is a **minimal DPFSA** for $p$ if it defines the same probability distribution as $p$ and there is no weakly equivalent DPFSA with fewer states.*

## 3 Recurrent Neural Language Models

**RNN LMs** are LMs whose conditional distributions are given by a recurrent neural network. We will focus on Elman RNNs (Elman, 1990) as they are the easiest to analyze and special cases of more common networks, e.g., those based on long short-term memory (LSTM; Hochreiter and Schmidhuber, 1997) and gated recurrent units (GRUs; Cho et al., 2014).

---

[6]Throughout the text, we use $\circ$ as a placeholder a free quantity, in this case, to any weight $w \in \mathbb{R}$. In case there are multiple $\circ$'s in an expression, they are not tied in any way.

**Definition 3.1.** *An **Elman RNN** (ERNN) $\mathcal{R} = (\Sigma, \sigma, D, \mathbf{U}, \mathbf{V}, \mathbf{b}, \mathbf{h}_0)$ is an RNN with the following hidden state recurrence:*

$$\mathbf{h}_t \stackrel{\text{def}}{=} \sigma\left(\mathbf{U}\mathbf{h}_{t-1} + \mathbf{V}\mathbf{r}(y_t) + \mathbf{b}\right), \qquad (2)$$

*where $\mathbf{h}_0$ is set to some vector in $\mathbb{R}^D$. $\mathbf{r}\colon \overline{\Sigma} \to \mathbb{R}^R$ is the symbol representation function and $\sigma$ is an element-wise nonlinearity. $\mathbf{b} \in \mathbb{R}^D$, $\mathbf{U} \in \mathbb{R}^{D \times D}$, and $\mathbf{V} \in \mathbb{R}^{D \times R}$. We refer to the dimensionality of the hidden state, $D$, as the **size** of the RNN.*

An RNN $\mathcal{R}$ can be used to specify an LM by using the hidden states to define the conditional distributions for $y_t$ given $\boldsymbol{y}_{<t}$.

**Definition 3.2.** *Let $\mathbf{E} \in \mathbb{R}^{|\overline{\Sigma}| \times D}$ and let $\mathcal{R}$ be an RNN. An **RNN LM** $(\mathcal{R}, \mathbf{E})$ is an LM whose conditional distributions are defined by projecting $\mathbf{E}\mathbf{h}_t$ onto the probability simplex $\boldsymbol{\Delta}^{|\Sigma|-1}$ using some $\mathbf{f}\colon \mathbb{R}^{|\Sigma|} \to \boldsymbol{\Delta}^{|\Sigma|-1}$:*

$$p(y_t \mid \boldsymbol{y}_{<t}) \stackrel{\text{def}}{=} \mathbf{f}\left(\mathbf{E}\mathbf{h}_t\right)_{y_t}. \qquad (3)$$

*We term $\mathbf{E}$ the **output matrix**.*

The most common choice for $\mathbf{f}$ is the **softmax** defined for $\mathbf{x} \in \mathbb{R}^D$ and $d \in \mathbb{Z}_D$ as

$$\text{softmax}(\mathbf{x})_d \stackrel{\text{def}}{=} \frac{\exp\left(x_d\right)}{\sum_{d'=1}^{D} \exp\left(x_{d'}\right)}. \qquad (4)$$

An important limitation of the softmax is that it results in a distribution with full support for all $\mathbf{x} \in \mathbb{R}^D$. However, one can achieve 0 probabilities by including *extended* real numbers $\overline{\mathbb{R}} \stackrel{\text{def}}{=} \mathbb{R} \cup \{-\infty, \infty\}$: Any element with $x_d = -\infty$ will result in $\text{softmax}(\mathbf{x})_d = 0$.

Recently, a number of alternatives to the softmax have been proposed. This paper uses the sparsemax function (Martins and Astudillo, 2016), which can output sparse distributions:

$$\text{sparsemax}(\mathbf{x}) \stackrel{\text{def}}{=} \underset{\mathbf{p} \in \boldsymbol{\Delta}^{D-1}}{\text{argmin}} ||\mathbf{p} - \mathbf{x}||_2^2. \qquad (5)$$

Importantly, $\text{sparsemax}(\mathbf{x}) = \mathbf{x}$ for $\mathbf{x} \in \boldsymbol{\Delta}^{D-1}$.

**On determinism.** Unlike PFSAs, Elman RNNs (and most other popular RNN architectures, such as the LSTM and GRU) implement inherently deterministic transitions between internal states. As we show shortly, certain types of Elman RNNs are at most as expressive as deterministic PFSAs, meaning that they can not represent non-determinizable PFSAs.

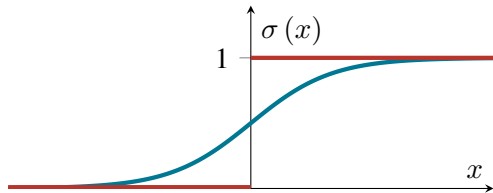

Figure 3: The sigmoid and Heaviside functions.

Common choices for the nonlinear function $\sigma$ in Eq. (2) are the sigmoid function $\sigma(x) = \frac{1}{1+\exp(-x)}$ and the ReLU $\sigma(x) = \max(0, x)$. However, the resulting nonlinear interactions of the parameters and the inputs make the analysis of RNN LMs challenging. One fruitful manner to make the analysis tractable is making a simplifying assumption about $\sigma$. We focus on a particularly useful simplification, namely the use of the Heaviside activation function.[7]

**Definition 3.3.** *The **Heaviside** function is defined as $H(x) \stackrel{\text{def}}{=} \mathbb{1}\{x > 0\}$.*

See Fig. 3 for the graph of the Heaviside function and its continuous approximation, the sigmoid. For cleaner notation, we define the set $\mathbb{B} \stackrel{\text{def}}{=} \{0, 1\}$. Using the Heaviside function, we can define the Heaviside ERNN, the main object of study in the rest of the paper.

**Definition 3.4.** *A **Heaviside Elman RNN** (HRNN) is an ERNN $\mathcal{R} = (\Sigma, \sigma, D, \mathbf{U}, \mathbf{V}, \mathbf{b}, \mathbf{h}_0)$ where $\sigma = H$.*

## 4 Equivalence of HRNNs and FSLMs

The hidden states of an HRNN live in $\mathbb{B}^D$, and can thus take $2^D$ different values. This invites an interpretation of $\mathbf{h}$ as the state of an underlying FSA that transitions between states based on the HRNN recurrence, specifying its local conditional distributions with the output matrix $\mathbf{E}$. Similarly, one can also imagine designing a HRNN that simulates the transitions of a given FSA by appropriately specifying the parameters of the HRNN. We explore this connection formally in this section and present the main technical result of the paper. The central result that characterizes the representational capacity HRNN can be informally summarized by the following theorem.

---

[7]While less common now due to its non-differentiability, the Heaviside function was the original activation function used in early work on artificial neural networks due to its close analogy to the firing of brain neurons (McCulloch and Pitts, 1943; Minsky, 1954; Kleene, 1956).

**Theorem 4.1** (Informal). *HRNN LMs are equivalent to DPFSAs.*

We split this result into the question of *(i)* how DPFSAs can simulate HRNN LMs and *(ii)* how HRNN LMs can simulate DPFSAs.

### 4.1 DPFSAs Can Simulate HRNNs

**Lemma 4.1.** *For any HRNN LM, there exists a weakly equivalent DPFSA.*

The proof closely follows the intuitive connection between the $2^D$ possible configurations of the RNN hidden state and the states of the strongly equivalent DPFSA. The outgoing transition weights of a state $q$ are simply the conditional probabilities of the transition symbols conditioned on the RNN hidden state represented by $q$.[8] This implies that HRNNs are at most as expressive as DPFSAs, and as a consequence, strictly less expressive as non-deterministic PFSAs. We discuss the implications of this in §6.

### 4.2 HRNNs Can Simulate DPFSAs

This section discusses the other direction of Theorem 4.1, showing that a general DPFSA can be simulated by an HRNN LM using a variant of the classic theorem originally due to Minsky (1954). We give the theorem a probabilistic twist, making it relevant to language modeling.

**Lemma 4.2.** *Let $\mathcal{A} = (\Sigma, Q, \delta, \lambda, \rho)$ be a DPFSA. Then, there exists a weakly equivalent HRNN LM whose RNN is of size $|\Sigma||Q|$.*

We describe the full construction of an HRNN LM simulating a given DPFSA in the next subsection. The full construction is described to showcase the mechanism with which the HRNN can simulate the transitions of a given FSA and give intuition on why this might, in general, require a large number of parameters in the HRNN. Many principles and constraints of the simulation are also reused in the discussion of the lower bounds on the size of the HRNN required to simulate the DPFSA.

#### 4.2.1 Weighted Minsky's Construction

For a DPFSA $\mathcal{A} = (\Sigma, Q, \delta, \lambda, \rho)$, we construct an HRNN LM $(\mathcal{R}, \mathbf{E})$ with $\mathcal{R} = (\Sigma, \sigma, D, \mathbf{U}, \mathbf{V}, \mathbf{b}, \mathbf{h}_0)$ defining the same distribution over $\Sigma^*$. The idea is to simulate the transition function $\delta$ with the Elman recurrence by appropriately setting $\mathbf{U}$, $\mathbf{V}$, and $\mathbf{b}$. The transition weights defining the stringsums are represented in $\mathbf{E}$.

Let $n\colon Q \times \Sigma \to \mathbb{Z}_{|Q||\Sigma|}$, $m\colon \Sigma \to \mathbb{Z}_{|\Sigma|}$, and $\overline{m}\colon \overline{\Sigma} \to \mathbb{Z}_{|\overline{\Sigma}|}$ bijections. We use $n$, $m$, and $\overline{m}$ to define the one-hot encodings $[\![\cdot]\!]$ of state–symbol pairs and of the symbols, i.e., we assume that $[\![q, y]\!]_d = \mathbb{1}\{d = n(q, y)\}$ and $[\![y]\!]_d = \mathbb{1}\{d = m(y)\}$ for $q \in Q$ and $y \in \Sigma$.

**HRNN's hidden states.** The hidden state $\mathbf{h}_t$ of $\mathcal{R}$ will represent the one-hot encoding of the current state $q_t$ of $\mathcal{A}$ at time $t$ together with the symbol $y_t$ upon reading which $\mathcal{A}$ entered $q_t$. Formally,

$$\mathbf{h}_t = [\![(q_t, y_t)]\!] \in \mathbb{B}^{|Q||\Sigma|}. \qquad (6)$$

There is a small caveat: How do we set the incoming symbol of $\mathcal{A}$'s initial state $q_\iota$? As we show later, the symbol $y_t$ in $\mathbf{h}_t = [\![(q_t, y_t)]\!]$ does not affect the subsequent transitions—it is only needed to determine the target of the current transition. Therefore, we can set $\mathbf{h}_0 = [\![(q_\iota, y)]\!]$ for any $y \in \Sigma$.

**Encoding the transition function.** The idea of defining $\mathbf{U}$, $\mathbf{V}$, and $\mathbf{b}$ is for the Elman recurrence to perform, upon reading $y_{t+1}$, element-wise conjunction between the representations of the children of $q_t$ and the representation of the states $\mathcal{A}$ can transition into after reading in $y_{t+1}$ from *any state*.[9] The former is encoded in the recurrence matrix $\mathbf{U}$, which has access to the current hidden state encoding $q_t$ while the latter is encoded in the input matrix $\mathbf{V}$, which has access to the one-hot representation of $y_{t+1}$. Conjoining the entries in those two representations will, due to the determinism of $\mathcal{A}$, result in a single non-zero entry: One representing the state which can be reached from $q_t$ (1st component) using the symbol $y_{t+1}$ (2nd component); see Fig. 4.

More formally, the recurrence matrix $\mathbf{U}$ lives in $\mathbb{B}^{|\Sigma||Q| \times |\Sigma||Q|}$. Each column $\mathbf{U}_{:, n(q,y)}$ represents the children of the state $q$ in the sense that the column contains 1's at the indices corresponding to the state–symbol pairs $(q', y')$ such that $\mathcal{A}$ transitions from $q$ to $q'$ after reading in the symbol $y'$. That is, for $q, q' \in Q$ and $y, y' \in \Sigma$, we define

$$U_{n(q',y'),n(q,y)} \stackrel{\text{def}}{=} \mathbb{1}\left\{ q_t \xrightarrow{y'/\circ} q' \in \delta \right\}. \qquad (7)$$

Since $y$ is free, each column is repeated $|\Sigma|$-times: Once for every $y \in \Sigma$—this is why, after entering the next state, the symbol used to enter it, in the case of the initial state, any incoming symbol can be chosen to set $\mathbf{h}_0$.

---

[8]The full proof is presented in Appendix A.

[9]See Fact A.1 in Appendix A.1 for a discussion of how an HRNN can implement the logical AND operation.

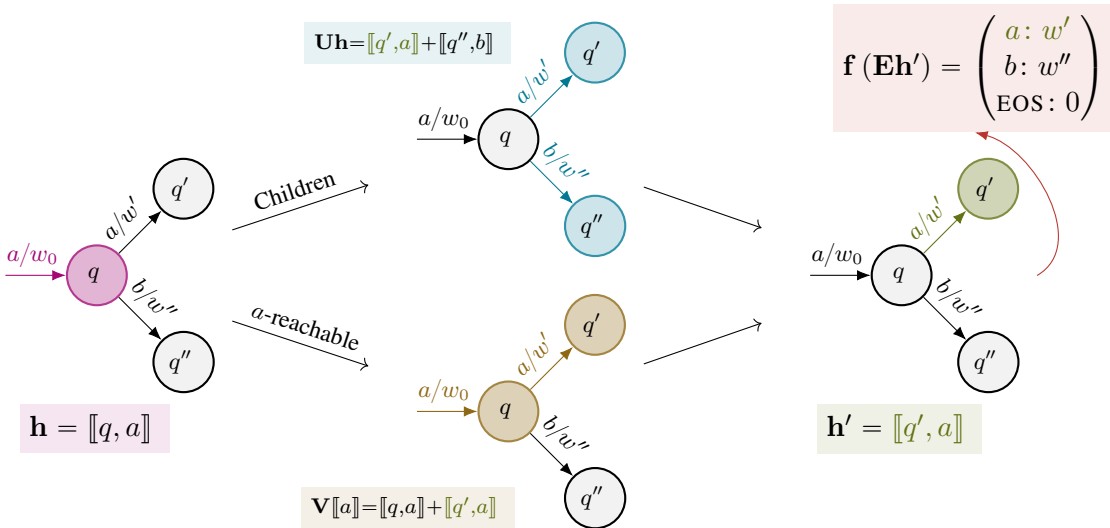

Figure 4: A high-level illustration of how the transition function of the FSA is simulated in Minsky's construction on a fragment of an FSA starting at $q$ (encoded in $\mathbf{h}$) and reading the symbol $a$. The top path disjoins the representations of the children of $q$, whereas the bottom path disjoins the representations of states reachable by an $a$-transition. The Heaviside activation conjoins these two representations into $\mathbf{h}'$ (rightmost fragment). Projecting $\mathbf{Eh}'$ results in the vector defining the same probability distribution as the outcoming arcs of $q$ (red box).

The input matrix $\mathbf{V}$ lives in $\mathbb{B}^{|\Sigma||Q| \times |\Sigma|}$ and encodes the information about which states can be reached by which symbols (from *any* state). The non-zero entries in the column corresponding to $y' \in \Sigma$ correspond to the state–symbol pairs $(q', y')$ such that $q'$ is reachable with $y'$ from *some* state:

$$V_{n(q',y'),m(y')} \overset{\text{def}}{=} \mathbb{1}\left\{ \circ \xrightarrow{y'/\circ} q' \in \delta \right\}. \quad (8)$$

Lastly, we define the bias as $\mathbf{b} \overset{\text{def}}{=} -\mathbf{1} \in \mathbb{R}^{|Q||\Sigma|}$, which allows the Heaviside function to perform the needed conjunction. The correctness of this process is proved in Appendix A (Lemma A.1).

**Encoding the transition probabilities.** We now turn to the second part of the construction: Encoding the string acceptance weights given by $\mathcal{A}$ into the probability distribution defined by $\mathcal{R}$. We present two ways of doing that: Using the standard softmax formulation, where we make use of the extended real numbers, and with the sparsemax.

The conditional probabilities assigned by $\mathcal{R}$ are controlled by the $|\overline{\Sigma}| \times |Q||\Sigma|$-dimensional output matrix $\mathbf{E}$. Since $\mathbf{h}_t$ is a one-hot encoding of the state–symbol pair $q_t, y_t$, the matrix–vector product $\mathbf{Eh}_t$ simply looks up the values in the $n(q_t, y_t)^{\text{th}}$ column. After being projected to $\mathbf{\Delta}^{|\overline{\Sigma}|-1}$, the entry in the projected vector corresponding to some $y_{t+1} \in \overline{\Sigma}$ should match the probability of $y_{t+1}$ given that $\mathcal{A}$ is in the state $q_t$, i.e., the weight on

the transition $q_t \xrightarrow{y_{t+1}/\circ} \circ$ if $y_{t+1} \in \Sigma$ and $\rho(q_t)$ if $y_{t+1} = \text{EOS}$. This is easy to achieve by simply encoding the weights of the outgoing transitions into the $n(q_t, y_t)^{\text{th}}$ column, depending on the projection function used. This is especially simple in the case of the sparsemax formulation. By definition, in a PFSA, the weights of the outgoing transitions and the final weight of a state $q_t$ form a probability distribution over $\overline{\Sigma}$ for every $q_t \in Q$. Projecting those values to the probability simplex, therefore, leaves them intact. We can therefore define

$$\mathbf{E}_{\overline{m}(y')n(q,y)} \overset{\text{def}}{=} \begin{cases} \tau(q \xrightarrow{y'/w} \circ) & | \text{ if } y' \in \Sigma \\ \rho(q) & | \text{ otherwise} \end{cases}.$$

$$(9)$$

Projecting the resulting vector $\mathbf{Eh}_t$, therefore, results in a vector whose entries represent the transition probabilities of the symbols in $\overline{\Sigma}$.

In the more standard softmax formulation, we proceed similarly but log the non-zero transition weights. Defining $\log 0 \overset{\text{def}}{=} -\infty$, we set

$$\mathbf{E}_{\overline{m}(y')n(q,y)} \overset{\text{def}}{=} \begin{cases} \log \tau(q \xrightarrow{y'/w} \circ) & | \text{ if } y' \in \Sigma \\ \log \rho(q) & | \text{ otherwise} \end{cases}.$$

$$(10)$$

It is easy to see that the entries of the vector $\text{softmax}(\mathbf{Eh}_t)$ form the same probability distribution as the original outgoing transitions out of $q$. Over the course of an entire input string, these weights are multiplied as the RNN transitions be-

tween different hidden states corresponding to the transitions in the original DPFSA $\mathcal{A}$. The proof can be found in Appendix A (Lemma A.2). This establishes the complete equivalence between HRNN LMs and FSLMs.[10]

# 5 Lower Bound on the Space Complexity of Simulating PFSAs with RNNs

Lemma 4.2 shows that HRNN LMs are at least as expressive as DPFSAs. More precisely, it shows that any DPFSA $\mathcal{A} = (\Sigma, Q, \delta, \lambda, \rho)$ can be simulated by an HRNN LM of size $\mathcal{O}(|Q||\Sigma|)$. In this section, we address the following question: How large does an HRNN LM have to be such that it can correctly simulate a DPFSA? We study the asymptotic bounds with respect to the size of the set of states, $|Q|$, as well as the number of symbols, $|\Sigma|$.

## 5.1 Asymptotic Bounds in $|Q|$

Intuitively, the $2^D$ configurations of a $D$-dimensional HRNN hidden state could represent $2^D$ states of a (DP)FSA. One could therefore hope to achieve exponential compression of a DPFSA by representing it as an HRNN LM.[11] Interestingly, this is not possible in general: Extending work by Dewdney (1977), Indyk (1995) shows that there exist unweighted FSAs which require an HRNN of size $\Omega\left(|\Sigma|\sqrt{|Q|}\right)$ to be simulated. At the same time, he also shows that *any* FSA can be simulated by an HRNN of size $\mathcal{O}\left(|\Sigma|\sqrt{|Q|}\right)$.[12]

We now ask whether the same lower bound can also be achieved when simulating DPFSAs. We find that the answer is negative: There exist DPFSAs which require an HRNN LM of size $\Omega(|\Sigma||Q|)$ to faithfully represent their probability distribution. Since the transition function of the underlying FSA can be simulated more efficiently, the bottleneck comes from the requirement of weak equivalence. Indeed, as the proof of the following theorem shows (Theorem 5.1 in Appendix A), the issue intuitively arises in the fact that, unlike in an HRNN LM, the local probability distributions of the different states in a PFSA are completely

---

[10]The full discussion of the result is postponed to §6.

[11]Indeed, any DPFSA defined from an RNN as described in the proof of Lemma 4.1 can naturally be exponentially compressed by representing it with an HRNN. However, not all DPFSAs are of this form.

[12]The constructions by Dewdney (1977) and Indyk (1995), which represent any unweighted FSA with a HRNN of size $\mathcal{O}\left(|\Sigma||Q|^{\frac{3}{4}}\right)$ and $\mathcal{O}\left(|\Sigma|\sqrt{|Q|}\right)$, respectively, are reviewed by Svete and Cotterell (2023).

arbitrary, whereas they are defined by shared parameters (the matrix $\mathbf{E}$) in an HRNN LM.

**Theorem 5.1.** *There exists a class of FSLMs $\{p_Q \mid Q = \{1, \ldots, N\}, N \in \mathbb{N}\}$ with minimal DPFSAs $\{\mathcal{A}_Q\}$ such that for every weakly equivalent HRNN LM to $p_Q$ and function $f(n) \in \omega(n)$ it holds that $D > f(|Q|)$.*

Note that the linear lower bound holds in the case that the transition matrix of the DPFSA, which corresponds to the output matrix $\mathbf{E}$ in the RNN LM, is full-rank. If the transition matrix is low-rank, its possible decomposition into smaller matrices could possibly be carried over to the output matrix of the RNN, reducing the size of the hidden state to the rank of the matrix.

## 5.2 Asymptotic Bounds in $|\Sigma|$

Since each of the input symbols can be encoded in $\log |\Sigma|$ bits, one could expect that the linear factor in the size of the alphabet from the constructions above could be reduced to $\mathcal{O}(\log |\Sigma|)$. However, we again find that such reduction is in general not possible—the set of FSAs presented in Appendix B is an example of a family that requires an HRNN whose size scales linearly with $|\Sigma|$ to be simulated correctly, which implies the following theorem.

**Theorem 5.2.** *There exists a class of FSLMs $\{p_\Sigma \mid \Sigma = \{y_1, \ldots, y_N\}, N \in \mathbb{N}\}$ such that for every weakly equivalent HRNN LM to $p_\Sigma$ and function $f(n) \in \omega(n)$ it holds that $D > f(|\Sigma|)$.*

Based on the challenges encountered in the example from Appendix B, we devise a simple sufficient condition for a logarithmic compression with respect to $|\Sigma|$ to be possible: Namely, that for any pair of states $q, q' \in Q$, there is at most a single transition leading from $q$ to $q'$. Importantly, this condition is met by classical $n$-gram LMs and by the languages studied by Hewitt et al. (2020). This intuitive characterization can be formalized by a property we call $\log |\Sigma|$-separability.

**Definition 5.1.** *An FSA $\mathcal{A} = (\Sigma, Q, I, F, \delta)$ is $\log |\Sigma|$-separable if it is deterministic and, for any pair $q, q' \in Q$, there is at most one symbol $y \in \Sigma$ such that $q \xrightarrow{y} q' \in \delta$.*

The conditional of $\log |\Sigma|$-separability is a relatively restrictive condition. To amend that, we introduce a simple procedure which, at the expense of enlarging the state space by a factor of $|\Sigma|$, transforms a general deterministic (unweighted) FSA into a $\log |\Sigma|$-separable one. Since this

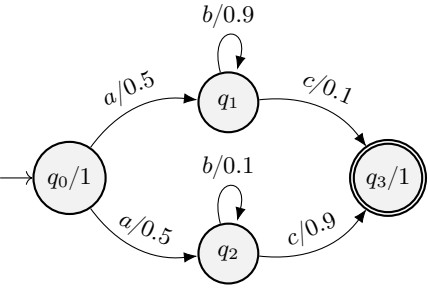

Figure 5: A non-determinizable PFSA. It assigns the string $ab^n c$ the probability $\mathcal{A}\left(ab^n c\right) = 0.5 \cdot 0.9^n \cdot 0.1 + 0.5 \cdot 0.1^n \cdot 0.9$, which can not be expressed as a single term for arbitrary $n \in \mathbb{N}_{\geqslant 0}$.

procedure does not apply to weighted automata, it is presented in Appendix C.

# 6 Discussion

In §4 and §5 we provided the technical results behind the relationship between HRNN LMs and DPFSAs. To put those results in context, we now discuss some of their implications.

## 6.1 Equivalence of HRNN LMs and DPFSAs

The equivalence between HRNN LMs and DPFSAs based on Lemmas 4.1 and 4.2 allows us to establish several constraints on the probability distributions expressible by HRNN LMs. For example, this result shows that HRNNs are at most as expressive as deterministic PFSAs and, therefore, *strictly less expressive* than general, non-deterministic, PFSAs due to the well-known result that not all non-deterministic PFSAs have a deterministic equivalent (Mohri, 1997).[13] An example of a simple non-determinizable PFSA, i.e., a PFSA whose distribution cannot be expressed by an HRNN LM, is shown in Fig. 5.[14]

Moreover, connecting HRNN LMs to DPFSAs allows us to draw on results from (weighted) formal language theory to manipulate and investigate HRNN LMs. For example, we can apply general results on the tightness of language models based on DPFSAs to HRNN LMs (Du et al., 2023, §5.1).[15]

Even if the HRNN LM is not tight *a priori*, the fact that the normalizing constant can be computed means that it can always be re-normalized to form a probability distribution over $\Sigma^*$. Furthermore, we can draw on the various results on the minimization of DPFSAs to reduce the size of the HRNN implementing the LM.

While Lemma 4.1 focuses on HRNN LMs and shows that they are finite-state, a similar argument could be made for any RNN whose activation functions map onto a finite set. This is the case with any RNN running on a computer with finite-precision arithmetic—in that sense, all deployed RNN LMs are finite-state, albeit with a very large state space. In other words, one can view RNNs as very compact representations of large DPFSAs whose transition functions are represented by the RNN's update function. Furthermore, since the topology and the weights of the implicit DPFSA are determined by the RNN's update function, the DPFSA can be learned very flexibly yet efficiently based on the training data. This is enabled by the sharing of parameters across the entire graph of the DPFSA instead of explicitly parametrizing every possible transition in the DPFSA or by hard-coding the allowed transitions as in $n$-gram LMs.

**A note on the use of the Heaviside function.** Minsky's construction uses the Heaviside activation function to implement conjunction. Note that, conveniently, we could also use the more popular $\mathrm{ReLU}$ function: A closer look at Minsky's construction shows that the only action performed by the Heaviside function is clipping negative values to $0$ while non-negative values are left intact.[16] Since $\mathrm{ReLU}$ behaves the same way on the relevant set of values, it could simply be swapped in for the Heaviside unit. This simply shows that the convenient binary structure of the Heaviside function does not enhance the representational capacity of the model in any way; as one would expect, ReLU-activated Elman RNN LMs are at least as expressive as Heaviside-activated ones.[17]

---

[13]General PFSAs are, in turn, equivalent to probabilistic regular grammars and discrete HMMs (Icard, 2020).

[14]Even if a non-deterministic PFSA can be determinized, the number of states of the determinized machine can be exponential in the size of the non-deterministic one (Buchsbaum et al., 2000). In this sense, non-deterministic PFSAs can be seen as exponentially compressed representations of FSLMs. The compactness of this non-deterministic representation must be "undone" using determinization before it can be encoded by an HRNN.

[15]Informally, the question of tightness concerns the question

of whether the LM forms a valid probability distribution over $\Sigma^*$, which is not necessarily the case for locally normalized LMs such as RNN LMs.

[16]More precisely, the only values that appear during the processing of a string are $-1$, $0$, and $1$, and the $-1$ is mapped to $0$ using the Heaviside function.

[17]Note that the same would be more difficult to say for sigmoid- or $\mathrm{tanh}$-activated Elman RNNs.

## 6.2 Space Complexity of Simulating DPFSAs with HRNN LMs

Theorems 5.1 and 5.2 establish lower bounds on how efficiently HRNN LMs can represent FSLMs, which are, to the best of our knowledge, the first results characterizing such space complexity. They reveal how the flexible local distributions of individual states in a PFSA require a large number of parameters in the simulating RNN to be matched. This implies that the simple Minsky's construction is in fact asymptotically *optimal* in the case of PFSAs, even though the transition function of the underlying FSA can be simulated more efficiently.

Nonetheless, the fact that RNNs can represent some FSLMs compactly is interesting. The languages studied by Hewitt et al. (2020) and Bhattamishra et al. (2020) can be very compactly represented by an HRNN LM and have clear linguistic motivations. Investigating whether other linguistically motivated phenomena in human language can be efficiently represented by HRNN LMs is an interesting area of future work, as it would yield insights into not only the full representational capacity of these models but also reveal additional inductive biases they use and that can be exploited for more efficient learning and modeling.

## 7 Related Work

To the best of our knowledge, the only existing connection between RNNs and weighted automata was made by Peng et al. (2018), where the authors connect the recurrences analogous to Eq. (2) of different RNN variants to the process of computing the probability of a string under a general PFSA. With this, they are able to show that the *hidden states* of an RNN can be used to store the probability of the input string, which can be used to upper-bound the representational capacity of specific RNN variants. Importantly, the interpretation of the hidden state is different to ours: Rather than tracking the current *state* of the PFSA, Peng et al. (2018)'s construction stores the *distribution* over all possible states. While this suggests a way of simulating PFSAs, the translation of the probabilities captured in the hidden state to the probability under an RNN LM is not straightforward.

Weiss et al. (2018), Merrill (2019) and Merrill et al. (2020) consider the representational capacity of *saturated RNNs*, whose parameters take their limiting values $\pm\infty$ to make the updates to the hidden states discrete. In this sense, their formal model is similar to ours. However, rather than considering the probabilistic representational capacity, they consider the flexibility of the update mechanisms of the variants in the sense of their long-term dependencies and the number of values the hidden states can take as a function of the string length. Connecting the assumptions of saturated activations with the results of Peng et al. (2018), they establish a hierarchy of different RNN architectures based on whether their update step is finite-state and whether the hidden state can be used to store arbitrary amounts of information. Analogous to our results, they show that Elman RNNs are finite-state while some other variants such as LSTMs are provably more expressive.

In a different line of work, Weiss et al. (2019) study the ability to *learn* a concise DPFSA from a given RNN LM. This can be seen as a relaxed setting of the proof of Lemma 4.1, where multiple hidden states are merged into a single state of the learned DPFSA to keep the representation compact. The work also discusses the advantages of considering *deterministic* models due to their interpretability and computational efficiency, motivating the connection between LMs and DPFSAs.

Discussion of some additional (less) related work can be found in Appendix D.

## 8 Conclusion

We prove that Heaviside Elman RNNs define the same set of probability distributions over strings as the well-understood class of deterministic probabilistic finite-state automata. To do so, we extend Minsky's classical construction of an HRNN simulating an FSA to the probabilistic case. We show that Minsky's construction is in some sense also optimal: Any HRNN representing the same distribution as some DPFSA over strings from an alphabet $\Sigma$ will, in general, require hidden states of size at least $\Omega\left(|\Sigma||Q|\right)$, which is the space complexity of Minsky's construction.

## Limitations

This paper aims to provide a *first step* at understanding modern LMs with weighted formal language theory and thus paints an incomplete picture of the entire landscape. While the formalization we choose here has been widely adopted in previous work (Minsky, 1954; Dewdney, 1977; Indyk, 1995), the assumptions about the models we make, e.g., binary activations and the simple recurrent

steps, are overly restrictive to represent the models used in practice; see also §6 for a discussion on the applicability to more complex models. It is likely that different formalizations of the RNN LM, e.g., those with asymptotic weights (Weiss et al., 2018; Merrill et al., 2020; Merrill, 2019) would yield different theoretical results. Furthermore, any inclusion of infinite precision would bring RNN LMs much higher up on the Chomsky hierarchy (Siegelmann and Sontag, 1992). Studying more complex RNN models, such as LSTMs, could also yield different results, as LSTMs are known to be in some ways more expressive than simple RNNs (Weiss et al., 2018; Merrill et al., 2020).

Another important aspect of our analysis is the use of explicit constructions to show the representational capacity of various models. While such constructions show theoretical equivalence, it is unlikely that trained RNN LMs would learn the proposed mechanisms in practice, as they tend to rely on dense representations of the context (Devlin et al., 2019). This makes it more difficult to use the results to analyze trained models. Rather, our results aim to provide theoretical upper bounds of what *could* be learned.

Lastly, we touch upon the applicability of finite-state languages to the analysis of human language. Human language is famously thought to not be finite-state (Chomsky, 1957), and while large portions of it might be modellable by finite-state machines, such formalisms lack the structure and interpretability of some mechanisms higher on the Chomsky hierarchy. For example, the very simple examples of (bounded) nesting expressible with context-free grammars are relatively awkward to express with finite-state formalisms such as finite-state automata—while they are expressible with such formalisms, the implementations lack the conciseness (and thus inductive biases) of the more concise formalisms. On the other hand, some prior work suggests that finding finite-state mechanisms could nonetheless be useful for understanding the inner workings of LMs and human language (Hewitt et al., 2020).

## Ethics Statement

The paper provides a way to theoretically analyze language models. To the best knowledge of the authors, there are no ethical implications of this paper.

## Acknowledgements

Ryan Cotterell acknowledges support from the Swiss National Science Foundation (SNSF) as part of the "The Forgotten Role of Inductive Bias in Interpretability" project. Anej Svete is supported by the ETH AI Center Doctoral Fellowship. We thank William Merrill for his thorough feedback on a draft of this paper as well as the students of the LLM course at ETH Zürich (263-5354-00L) for carefully reading an early version of this paper as part of their lecture notes.

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

## A Proofs

### A.1 Performing the Logical AND with an HRNN

Minsky's construction requires the RNN to perform the logical AND operation between specific entries of binary vectors $\mathbf{x} \in \mathbb{B}^D$. The following fact shows how this can easily be performed by an HRNN with appropriately set parameters.

**Fact A.1.** *Consider $m$ indices $i_1, \ldots, i_m \in \mathbb{Z}_D$ and vectors $\mathbf{x}, \mathbf{v} \in \mathbb{B}^D$ such that $v_i = \mathbb{1}\{i \in \{i_1, \ldots, i_m\}\}$, i.e., with entries $1$ at indices $i_1, \ldots, i_m$. Then, $H\left(\mathbf{v}^\top \mathbf{x} - (m-1)\right) = 1$ if and only if $x_{i_k} = 1$ for all $k = 1, \ldots, m$. In other words,*

$$H\left(\mathbf{v}^\top \mathbf{x} - (m-1)\right) = x_{i_1} \wedge \cdots \wedge x_{i_m}. \tag{11}$$

As a special case, $m = 2$ in Fact A.1 corresponds to the AND operation of two elements, which is used in Minsky's construction. There, the vector $\mathbf{v}$ corresponds to the weights of a single neuron while $-(m-1)$ ($-1$ in case $m = 2$) corresponds to its bias.

We now present the proofs of the lemmas establishing the equivalence of DPFSAs and HRNN LMs.

**Lemma 4.1.** *For any HRNN LM, there exists a weakly equivalent DPFSA.*

*Proof.* Let $\mathcal{R} = (\Sigma, \sigma, D, \mathbf{U}, \mathbf{V}, \mathbf{b}, \mathbf{h}_0)$ be a HRNN defining the locally normalized language model $p$. We construct a weakly equivalent DPFSA $\mathcal{A} = (\Sigma, Q, \delta, \lambda, \rho)$. Construct a bijection $s \colon \mathbb{B}^D \to \mathbb{Z}_{2^D}$. Now, for every state $q \stackrel{\text{def}}{=} s(\mathbf{h}) \in Q \stackrel{\text{def}}{=} \mathbb{Z}_{2^D}$, construct a transition $q \xrightarrow{y/w} q'$ where $q' = s(\sigma\left(\mathbf{U}\mathbf{h} + \mathbf{V}[\![y]\!] + \mathbf{b}\right))$ with the weight $w = p\left(y \mid \mathbf{h}\right) = \mathbf{f}\left(\mathbf{E}\mathbf{h}\right)_y$. We define the initial function as $\lambda\left(s(\mathbf{h})\right) = \mathbb{1}\{\mathbf{h} = \mathbf{h}_0\}$ and final function $\rho$ with $\rho\left(q\right) \stackrel{\text{def}}{=} p\left(\text{EOS} \mid s(q)\right)$. It is easy to see that $\mathcal{A}$ defined this way is deterministic. We now prove that the weights assigned to strings by $\mathcal{A}$ and $\mathcal{R}$ are the same. Define $q_0 \stackrel{\text{def}}{=} s(\mathbf{h}_0)$ and let $\mathbf{y} \in \Sigma^*$ with $|\mathbf{y}| = T$. Then, let

$$\boldsymbol{\pi} = \left(q_0 \xrightarrow{y_1/w_1} q_1, \ldots, q_{T-1} \xrightarrow{y_T/w_T} q_T\right). \tag{12}$$

be the path with the scan $\mathbf{y}$ and starting in $q_0$ (such a path exists since we the defined automaton is *complete*—all possible transitions are defined for all states). Then, it holds that

$$
\begin{aligned}
\mathcal{A}\left(\mathbf{y}\right) &= \lambda\left(q_0\right) \cdot \left[\prod_{t=1}^{T} w_t\right] \cdot \rho\left(q_T\right) \\
&= 1 \cdot \prod_{t=1}^{T} p\left(y_t \mid s^{-1}(q_t)\right) \cdot p\left(\text{EOS} \mid s^{-1}(q_T)\right) \\
&= p\left(\mathbf{y}\right)
\end{aligned}
$$

which is exactly the weight assigned to $\mathbf{y}$ by $\mathcal{R}$. Note that all paths not starting in $s(\mathbf{h}_0)$ have weight $0$ due to the definition of the initial function. ∎

**Lemma A.1.** *Let $\mathcal{A} = (\Sigma, Q, \delta, \lambda, \rho)$ be a deterministic PFSA, $\mathbf{y} = y_1 \ldots y_T \in \Sigma^*$, and $q_t$ the state arrived at by $\mathcal{A}$ upon reading the prefix $\mathbf{y}_{\leqslant t}$. Let $\mathcal{R}$ be the HRNN specified by the Minsky construction for $\mathcal{A}$, $n$ the permutation defining the one-hot representations of state-symbol pairs by $\mathcal{R}$, and $\mathbf{h}_t$ $\mathcal{R}$'s hidden state after reading $\mathbf{y}_{\leqslant t}$. Then, it holds that $\mathbf{h}_0 = [\![(q_\iota, y)]\!]$ where $q_\iota$ is the initial state of $\mathcal{A}$ and $y \in \Sigma$ and $\mathbf{h}_T = [\![(q_T, y_T)]\!]$.*

*Proof.* Define $s(\mathbf{h} = [\![(q, y)]\!]) \stackrel{\text{def}}{=} q$. We can then restate the lemma as $s(\mathbf{h}_T) = q_T$ for all $\mathbf{y} \in \Sigma^*$, $|\mathbf{y}| = T$. Let $\boldsymbol{\pi}$ be the $\mathbf{y}$-labeled path in $\mathcal{A}$. We prove the lemma by induction on the string length $T$.

**Base case: $T = 0$.** Holds by the construction of $\mathbf{h}_0$.

**Inductive step:** $T > 0$**.** Let $\boldsymbol{y} \in \Sigma^*$ with $|\boldsymbol{y}| = T$ and assume that $s(\mathbf{h}_{T-1}) = q_{T-1}$. We prove that the specifications of $\mathbf{U}$, $\mathbf{V}$, and $\mathbf{b}$ ensure that $s(\mathbf{h}_T) = q_T$. By definition of the recurrence matrix $\mathbf{U}$ (cf. Eq. (7)), the vector $\mathbf{U}\mathbf{h}_{T-1}$ will contain a 1 at the entries $n\left(q', y'\right)$ for $q' \in Q$ and $y' \in \Sigma$ such that $q_T \xrightarrow{y'/\circ} q' \in \delta$. This can equivalently be written as $\mathbf{U}\mathbf{h}_{T-1} = \bigvee_{q_T \xrightarrow{y'/\circ} q' \in \delta} [\![(q', y')]\!]$, where the disjunction is applied element-wise.

On the other hand, by definition of the input matrix $\mathbf{V}$ (cf. Eq. (8)), the vector $\mathbf{V}[\![y_T]\!]$ will contain a 1 at the entries $n\left(q', y_T\right)$ for $q' \in Q$ such that $\circ \xrightarrow{y_T/\circ} q' \in \delta$. This can also be written as $\mathbf{V}[\![y_T]\!] = \bigvee_{\circ \xrightarrow{y_T/\circ} q' \in \delta} [\![(q', y_T)]\!]$.

By Fact A.1, $H\left(\mathbf{U}\mathbf{h}_{T-1} + \mathbf{V}[\![y_T]\!] + \mathbf{b}\right)_{n(q', y')} = H\left(\mathbf{U}\mathbf{h}_{T-1} + \mathbf{V}[\![y_T]\!] - \mathbf{1}\right)_{n(q', y')} = 1$ holds if and only if $\left(\mathbf{U}\mathbf{h}_{T-1}\right)_{n(q', y')} = 1$ and $\left(\mathbf{V}[\![y_T]\!]\right)_{n(q', y')} = 1$. This happens if

$$q_T \xrightarrow{y'/\circ} q' \in \delta \text{ and } \circ \xrightarrow{y_T/\circ} q' \in \delta \iff q_T \xrightarrow{y_T/\circ} q', \tag{13}$$

i.e., if and only if $\mathcal{A}$ transitions from $q_T$ to $q_T$ upon reading $y_T$ (it transitions only to $q_T$ due to determinism).

Since the string $\boldsymbol{y}$ was arbitrary, this finishes the proof. ∎

**Lemma A.2.** *Let* $\mathcal{A} = (\Sigma, Q, \delta, \lambda, \rho)$ *be a deterministic PFSA,* $\boldsymbol{y} = y_1 \ldots y_T \in \Sigma^*$, *and* $q_t$ *the state arrived at by* $\mathcal{A}$ *upon reading the prefix* $\boldsymbol{y}_{\leqslant t}$. *Let* $\mathcal{R}$ *be the HRNN specified by the Minsky construction for* $\mathcal{A}$, $\mathbf{E}$ *the output matrix specified by the generalized Minsky construction,* $n$ *the permutation defining the one-hot representations of state-symbol pairs by* $\mathcal{R}$, *and* $\mathbf{h}_t$ $\mathcal{R}$*'s hidden state after reading* $\boldsymbol{y}_{\leqslant t}$. *Then, it holds that* $p\left(\boldsymbol{y}\right) = \mathcal{A}\left(\boldsymbol{y}\right)$.

*Proof.* Let $\boldsymbol{y} \in \Sigma^*$, $|\boldsymbol{y}| = T$ and let $\boldsymbol{\pi}$ be the $\boldsymbol{y}$-labeled path in $\mathcal{A}$. Again, let $\overline{p}\left(\boldsymbol{y}\right) \stackrel{\text{def}}{=} \prod_{t=1}^{|\boldsymbol{y}|} p\left(y_t \mid \boldsymbol{y}_{<t}\right)$. We prove $\overline{p}\left(\boldsymbol{y}\right) = \prod_{t=1}^T w_t$ by induction on $T$.

**Base case:** $T = 0$**.** In this case, $\boldsymbol{y} = \varepsilon$, i.e., the empty string, and $\mathcal{A}\left(\varepsilon\right) = 1$. $\mathcal{R}$ computes $\overline{p}\left(\varepsilon\right) = \prod_{t=1}^0 p\left(y_t \mid \boldsymbol{y}_{<t}\right) = 1$.

**Inductive step:** $T > 0$**.** Assume that the $\overline{p}\left(y_1 \ldots y_{T-1}\right) = \prod_{t=1}^{T-1} w_t$. By Lemma A.1, we know that $s(\mathbf{h}_{T-1}) = q_T$ and $s(\mathbf{h}_T) = q_T$. By the definition of $\mathbf{E}$ for the specific $\mathbf{f}$, it holds that $\mathbf{f}\left(\mathbf{E}\mathbf{h}_{T-1}\right)_{m(y)} = \tau(s(\mathbf{h}_{T-1}) \xrightarrow{y/w_T} s(\mathbf{h}_T)) = w_T$. This means that $\overline{p}\left(\boldsymbol{y}_{\leqslant T}\right) = \prod_{t=1}^T w_t$, which is what we wanted to prove.

Clearly, $p\left(\boldsymbol{y}\right) = \overline{p}\left(\boldsymbol{y}\right) p\left(\text{EOS} \mid \boldsymbol{y}\right)$. By the definition of $\mathbf{E}$ (cf. Eq. (9)), $\left(\mathbf{E}\mathbf{h}_T\right)_{m(\text{EOS})} = \rho\left(s(\mathbf{h}_T)\right)$, meaning that $p\left(\boldsymbol{y}\right) = \overline{p}\left(\boldsymbol{y}\right) p\left(\text{EOS} \mid \boldsymbol{y}\right) = \prod_{t=1}^T w_t \rho\left(s(\mathbf{h}_T)\right) = \mathcal{A}\left(\boldsymbol{y}\right)$. Since $\boldsymbol{y} \in \Sigma^*$ was arbitrary, this finishes the proof.

∎

**A note on strong equivalence.** The purpose of Lemma A.2 and Lemma 4.1 was to show the existence of a weakly equivalent (cf. Definition 2.1) HRNN LM given a DPFSA defining a finite-state LM and vice versa. We keep the discussion in the main part of the paper restricted to weak equivalence for brevity. However, note that the proofs of the lemmas in fact establish the existence of a *strongly* equivalent DPFSA and HRNN LM, respectively. This can easily be seen from the one-to-one correspondence between path scanning a given string in the DPFSA and the sequence of hidden states generating the same string in the HRNN LM. In this sense, the connection between DPFSAs and HRNN LMs is even tighter than just defining the same probability distribution; however, we are mainly interested in the implications of the simpler weak equivalence.

**Theorem 5.1.** *There exists a class of FSLMs* $\{p_Q \mid Q = \{1, \ldots, N\}, N \in \mathbb{N}\}$ *with minimal DPFSAs* $\{\mathcal{A}_Q\}$ *such that for every weakly equivalent HRNN LM to* $p_Q$ *and function* $f\left(n\right) \in \omega\left(n\right)$ *it holds that* $D > f\left(|Q|\right)$.

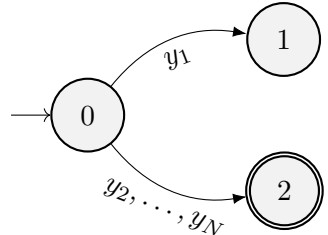

Figure 6: The FSA $\mathcal{A}_N$.

*Proof.* Without loss of generality, we work with $\overline{\mathbb{R}}$-valued hidden states. Let $\mathcal{A}$ be a minimal deterministic PFSA and $\mathcal{R} = (\Sigma, \sigma, D, \mathbf{U}, \mathbf{V}, \mathbf{b}, \mathbf{h}_0)$ a HRNN with $p(\boldsymbol{y}) = \mathcal{A}(\boldsymbol{y})$ for every $\boldsymbol{y} \in \Sigma^*$. Let $\boldsymbol{y}_{<T} \in \Sigma^*$ and $\boldsymbol{y}_{\leqslant T} \stackrel{\text{def}}{=} \boldsymbol{y}_{<T} y$ for some $y \in \Sigma$. Define $\overline{p}(\boldsymbol{y}) \stackrel{\text{def}}{=} \prod_{t=1}^{|\boldsymbol{y}|} p(y_t \mid \boldsymbol{y}_{<t})$. It is easy to see that $\overline{p}(\boldsymbol{y}_{<T} y_T) = \overline{p}(\boldsymbol{y}_{<T}) p(y_t \mid \boldsymbol{y}_{<T})$. The probabilities in the conditional distribution $p(\cdot \mid \boldsymbol{y}_{<T})$ are determined by the values in $\mathbf{E}\mathbf{h}_{T-1}$. By definition of the deterministic PFSA, there are $|Q|$ such conditional distributions. Moreover, these distributions (represented by vectors $\in \boldsymbol{\Delta}^{|\Sigma|-1}$) can generally be *linearly independent*.[18] This means that for any $q$, the probability distribution of the outgoing transitions can not be expressed as a linear combination of the probability distributions of other states. To express the probability vectors for all states, the columns of the output matrix $\mathbf{E}$, therefore, have to span $\overline{\mathbb{R}}^{|Q|}$, implying that $\mathbf{E}$ must have at least $|Q|$ columns. This means that the total space complexity (and thus the size of the HRNN representing the same distribution as $\mathcal{A}$) is $\Omega(|Q|)$. ∎

## B Lower Space Bounds in $|\Sigma|$ for Simulating Deterministic PFSAs with HRNNs

In this section, we provide a family of DPFSAs which require a HRNN LM whose size must scale linearly with the size of the alphabet. We also provide a sketch of the proof of why a compression in $|\Sigma|$ is not possible. Let $\mathcal{A}_N = (\Sigma_N, \{0, 1\}, \{0\}, \{1\}, \delta_N)$ be an FSA over the alphabet $\Sigma_N = \{y_1, \ldots, y_N\}$ such that $\delta_N = \left\{ 0 \xrightarrow{y_1} 1 \right\} \cup \left\{ 0 \xrightarrow{y_n} 2 \mid n = 2, \ldots N \right\}$ (see Fig. 6).

Clearly, to be able to correctly represent all local distributions of the DPFSA, the HRNN LM must contain a representation of each possible state of the DPFSA in a unique hidden state. On the other hand, the only way that the HRNN can take into account the information about the current state $q_t$ of the simulated FSA $\mathcal{A}$ is through the hidden state $\mathbf{h}_t$. The hidden state, in turn, only interacts with the recurrence matrix $\mathbf{U}$, which does not have access to the current input symbol $y_{t+1}$. The only interaction between the current state and the input symbol is thus through the addition in $\mathbf{U}\mathbf{h}_t + \mathbf{V}[\![y_{t+1}]\!]$. This means that, no matter how the information about $q_t$ is encoded in $\mathbf{h}_t$, to be able to take into account all possible transitions stemming in $q_t$ (before taking into account $y_{t+1}$), $\mathbf{U}\mathbf{h}_t$ must activate *all* possible next states, i.e., all children of $q_t$. On the other hand, since $\mathbf{V}[\![y_{t+1}]\!]$ does not have precise information about $q_t$, it must activate all states which can be entered with an $y_{t+1}$-transition, just like in Minsky's construction.

In Minsky's construction, the recognition of the correct next state was done by keeping a separate entry (one-dimensional sub-vector) for each possible pair $q_{t+1}, y_{t+1}$. However, when working with compressed representations of states (e.g., in logarithmic space), a single common sub-vector of size $< |\Sigma|$ (e.g., $\log |\Sigma|$) has to be used for all possible symbols $y \in \Sigma$. Nonetheless, the interaction between $\mathbf{U}\mathbf{h}_t$ and $\mathbf{V}[\![y_{t+1}]\!]$ must then ensure that only the correct state $q_{t+1}$ is activated. For example, in Minsky's construction, this was done by simply taking the conjunction between the entries corresponding to $q, y$ in $\mathbf{U}\mathbf{h}_t$ and the entries corresponding to $q', y'$ in $\mathbf{V}[\![y']\!]$, which were all represented in individual entries of the vectors. On the other hand, in the case of the $\log$ encoding, this could intuitively be done by trying to match the $\log |\Sigma|$ ones in the representation $(\mathbf{p}(y) \mid \mathbf{1} - \mathbf{p}(y))$, where $\mathbf{p}(y)$ represent the binary encoding of $y$. If the $\log |\Sigma|$ ones match (which is checked simply as it would result in a large enough sum in the corresponding entry of the matrix-vector product), the correct transition could be chosen (to perform the conjunction from Fact A.1 correctly, the bias would simply be set to $\log |\Sigma| - 1$). However,

---

[18]For this to be the case, it has to hold that $|\Sigma| \geqslant |Q|$.

an issue arises as soon as *multiple* dense representations of symbols in $\mathbf{V}[\![y]\!]$ have to be activated against the same sub-vector in $\mathbf{U}\mathbf{h}_t$—the only way this can be achieved is if the sub-vector in $\mathbf{U}\mathbf{h}_t$ contains the disjunction of the representations of all the symbols which should be activated with it. If this sets too many entries in $\mathbf{U}\mathbf{h}_t$ to one, this can result in "false positives". This is explained in more detail for the DPFSAs in Fig. 6 next.

Let $\mathbf{r}_n$ represent any dense encoding of $y_n$ in the alphabet of $\mathcal{A}_N$ (e.g., in the logarithmic case, that would be $(\mathbf{p}(n) \mid \mathbf{1} - \mathbf{p}(n))$). Going from the intuition outlined above, any HRNN simulating $\mathcal{A}_N$, the vector $\mathbf{U}\mathbf{h}_0$ must, among other things, contain a sub-vector corresponding to the states 1 and 2. The sub-vector corresponding to the state 2 must activate (through the interaction in the Heaviside function) against any $y_n$ for $n = 2, \ldots, N$ in $\mathcal{A}_N$. This means it has to match all representations $\mathbf{r}_n$ for all $n = 2, \ldots, N$. The only way this can be done is if the pattern for recognizing state 2 being entered with any $y_n$ for $n = 2, \ldots, N$ is of the form $\mathbf{r} = \bigvee_{n=2}^{N} \mathbf{r}_n$. However, for sufficiently large $N$, $\mathbf{r} = \bigvee_{n=2}^{N} \mathbf{r}_n$ will be a vector of all ones—including all entries active in $\mathbf{r}_1$. This means that *any* encoding of a symbol will be activated against it—among others, $y_1$. Upon reading $y_1$ in state 1, the network will therefore not be able to deterministically activate only the sub-vector corresponding to the correct state 1. This means that the linear-size encoding of the symbols is, in general, optimal for representing DPFSAs with HRNN LMs.

## C  Transforming a General Deterministic FSA into a $\log|\Sigma|$-separable FSA

$\log|\Sigma|$-separability is a relatively restrictive condition. To amend that, we introduce a simple procedure which, at the expense of enlarging the state space by a factor of $\Sigma$, transforms a general deterministic FSA into a $\log|\Sigma|$-separable one. We call this $\log|\Sigma|$-**separation**. Intuitively, it augments the state space by introducing a new state $(q, y)$ for every outgoing transition $q \xrightarrow{y} q'$ of every state $q \in Q$, such that $(q, y)$ simulates the only state the original state $q$ would transition to upon reading $y$. Due to the determinism of the original FSA, this results in a $\log|\Sigma|$-separable FSA with at most $|Q||\Sigma|$ states.

While the increase of the state space might seem like a step backward, recall that using Indyk's construction, we can construct an HRNN simulating an FSA whose size scales with the square root of the number of states. And, since the resulting FSA is $\log|\Sigma|$-separable, we can reduce the space complexity with respect to $\Sigma$ to $\log|\Sigma|$. This is summarized in the following theorem, which characterizes how compactly general deterministic FSAs can be encoded by HRNNs. To our knowledge, this is the tightest bound on simulating general unweighted deterministic FSAs with HRNNs.

**Theorem C.1.** *Let $\mathcal{A} = (\Sigma, Q, I, F, \delta)$ be a minimal FSA recognizing the language L. Then, there exists an HRNN $\mathcal{R} = (\Sigma, \sigma, D, \mathbf{U}, \mathbf{V}, \mathbf{b}, \mathbf{h}_0)$ accepting L with $D = \mathcal{O}\left(\log|\Sigma|\sqrt{|\Sigma||Q|}\right)$.*

The full $\log|\Sigma|$-separation procedure is presented in Algorithm 1. It follows the intuition of creating a separate "target" for each transition $q \xrightarrow{y} q'$ for every state $q \in Q$. To keep the resulting FSA deterministic, a new, artificial, initial state with no incoming transitions is added and is connected with the augmented with the children of the original initial state.

The following simple lemmata show the formal correctness of the procedure and show that it results in a $\log|\Sigma|$-separable FSA, which we need for compression in the size of the alphabet.

**Lemma C.1.** *For any $y \in \Sigma$, $(q, y) \xrightarrow{y'} (q', y') \in \delta'$ if and only if $q \xrightarrow{y'} q' \in \delta$.*

*Proof.* Ensured by the loop on Line 3. ∎

**Lemma C.2.** *$\log|\Sigma|$-separation results in an equivalent FSA.*

*Proof.* We have to show that, for any $\boldsymbol{y} \in \Sigma^*$, $\boldsymbol{y}$ leads to a final state in $\mathcal{A}$ if and only if $\boldsymbol{y}$ leads to a final state in $\mathcal{A}'$. For the string of length 0, this is clear by Lines 13 and 14. For strings of length $\geqslant 1$, it follows from Lemma C.1 that $\boldsymbol{y}$ leads to a state $q$ in $\mathcal{A}$ if and only if $\exists y \in \Sigma$ such that $\boldsymbol{y}$ leads to $(q, y)$ in $\mathcal{A}'$. From Lines 11 and 12, $(q, y) \in F'$ if and only if $q \in F$, finishing the proof. ∎

**Lemma C.3.** *$\log|\Sigma|$-separation results in a $\log|\Sigma|$-separable FSA.*

*Proof.* Since the state $(q', y')$ is the only state in $Q'$ transitioned to from $(q, y)$ after reading $y'$ (for any $y \in \Sigma$), it is easy to see that $\mathcal{A}'$ is indeed $\log|\Sigma|$-separable. ∎

**Algorithm 1**

---

1. **def** SEPARATE($\mathcal{A} = (\Sigma, Q, I, F, \delta)$):
2.     $\mathcal{A}' \leftarrow (\Sigma, Q' = Q \times \Sigma \cup \{q_\iota'\}, \delta' = \varnothing, I' = \{q_\iota'\}, F' = \varnothing)$
3.     ▷ *Connect the children of the original initial state $q_\iota$ with the new, aritificial, initial state.*
4.     **for** $y \in \Sigma$ :
5.         **for** $q_\iota \xrightarrow{y'} q' \in \delta$ :
6.             **add** $q_\iota' \xrightarrow{y} (q', y')$ **to** $\delta'$
7.     **for** $q \in Q, y \in \Sigma$ :
8.         **for** $q \xrightarrow{y'} q' \in \delta$ :
9.             **add** $(q, y) \xrightarrow{y'} (q', y')$ **to** $\delta'$
10.    ▷ *Add all state-symbol pairs with a state from the original set of final states to the new set of final states.*
11.    **for** $q_\varphi \in F, y \in \Sigma$ :
12.        **add** $(q_\varphi, y)$ **to** $F'$
13.    **if** $q_\iota \in I$ :        ▷ *Corner case: If the original initial state $q_\iota$ is an initial state, make the artificial initial state $q_\iota'$ final.*
14.        **add** $q_\iota'$ **to** $F'$
15.    **return** $\mathcal{A}'$

---

## D  Additional Related Work

Our work characterizes the representational capacity of HRNN LMs in terms of DPFSAs. On the other end of representational capacity, Chen et al. (2018); Nowak et al. (2023) consider the connection between Elman RNNs with arbitrary precision—a stark contrast to our model—and (probabilistic) Turing machines first established by Siegelmann and Sontag (1992). They outline some implications the relationship has on the representational capacity of RNNs and the solvability of tasks such as finding the most probable string or deciding whether an RNN is tight. These tasks are shown to be undecidable. This is in contrast to the equivalence shown here which, among other things, means that the decidability of the tasks on PFSAs can be carried over to RNN LMs.

On a different note, Bhattamishra et al. (2020) and Deletang et al. (2023) provide an *empirical* survey of the unweighted representational capacity of different LM architectures. The former focuses on RNN variants and their ability to recognize context-free languages. The authors find that RNNs indeed struggle to learn the mechanisms required to recognize context-free languages, but find that hierarchical languages of finite depth, such as $\mathrm{D}(k, m)$, can be learned reliably. This further motivates the connection between RNN LMs and finite-state models, as well as the specific construction by Hewitt et al. (2020). While the results from Deletang et al. (2023) can be connected to the theoretical insights provided by existing work, it is also clear that the probabilistic nature, as well as non-architectural aspects of LMs (such as the training regime), make establishing a clear hierarchy of models difficult.[19]

---

[19] The hierarchy of probabilistic formal languages is not as clear as the original Chomsky hierarchy, which might be one of the reasons behind the inconsistent results (Icard, 2020).