# OpenReview forum: "Recurrent Neural Language Models as Probabilistic Finite-state Automata"
_EMNLP/2023/Conference — EMNLP 2023 Main_

### Official Review · Reviewer_bmNn · 2023-08-04

**Typos Grammar Style And Presentation Improvements:** none
**Soundness:** 5

**Excitement:**

4: Strong: This paper deepens the understanding of some phenomenon or lowers the barriers to an existing research direction.

**Missing References:**

none

**Paper Topic And Main Contributions:**

The aim of this paper is studying the expressive power of recurrent neural networks as language models, that is considering the probability distribution they associate to strings. The conclusion is that they associate the same probability distribution as deterministic probabilistic finite-state automata, but only when some simplifying assumptions are considered, including the Heaviside activation function. Space complexity is then extensively studied for the simulation of PFSAs with RNNs.

The paper only reports the results, leaving some discussions in addition to proofs to the appendixes, which are then very long.

**Questions For The Authors:**

none

**Reasons To Accept:**

--- interesting study and results

--- paper well organized and clearly written

**Reasons To Reject:**

--- a lot of the material had to be moved to the appendixes

**Reproducibility:**

N/A: Doesn't apply, since the paper does not include empirical results.

**Reviewer Confidence:**

3: Pretty sure, but there's a chance I missed something. Although I have a good feel for this area in general, I did not carefully check the paper's details, e.g., the math, experimental design, or novelty.

---

> ### Author Rebuttal · Authors · 2023-08-27
>
> We thank the reviewer for the review and for recognizing the utility of the work and for the nice words about the clarity.
>
> We are grateful for the feedback about the appendices and the discussion of the results.
> We agree that the appendices of the paper are long, however, we would like to point out that most of the space is taken up by the review of the asymptotically more efficient constructions (due to Dewdney and Indyk) not directly necessary for the understanding of the main contributions of the work. Nonetheless, we agree that the discussion of the results is important and should not be relegated to the appendix. Therefore, should the paper be accepted, we plan to use part of the additional space to move the discussion into the main part of the paper and incorporate it there. We also think this will frame the discussion better.
>
> We also sincerely thank the reviewer for the high numeric scores awarded to the submission. Nonetheless, we would like to ask about the soundness score—is anything in the exposition not clear or convincing? If so, we would be happy to further clarify and accordingly improve the submission.

---

### Official Review · Reviewer_uUpK · 2023-08-04

**Typos Grammar Style And Presentation Improvements:** N/A
**Soundness:** 5

**Excitement:**

4: Strong: This paper deepens the understanding of some phenomenon or lowers the barriers to an existing research direction.

**Missing References:**

N/A

**Paper Topic And Main Contributions:**

The paper investigates the expressive power of Heaviside-activated Elman RNNs, and finds they define e the same set of probability distributions over strings as dPFSA.

**Questions For The Authors:**

line 308: A common simplification is the use of the Heaviside activation function --> Could the authors provide some citations to support the claim of "common simplification"?

**Reasons To Accept:**

1. Framing the study of RNN LMs through the lens of weighted formal language theory to make precise statements about their capabilities.
2. The paper analyzed the space complexity of simulating dPFSAs with HRNN LMs, proving linear lower bounds in both the number of states and the alphabet size. This reveals the limitations of Heaviside-activated Elman RNNs.

**Reasons To Reject:**

1. The Heaviside-activated Elman RNNs makes very strong simplifying assumptions about architecture (which is fine) and the activations (which makes the proof a little trivial, since the activations are binary variables) compared to LMs used in practice. As recognized by the authors in the Limitations section, most of the useful LMs (including word2vec) use dense representations.
2. Other limitations are also well-discussed in the Limitation Section. But overall, I think this is a good paper.

**Reproducibility:**

N/A: Doesn't apply, since the paper does not include empirical results.

**Reviewer Confidence:**

2: Willing to defend my evaluation, but it is fairly likely that I missed some details, didn't understand some central points, or can't be sure about the novelty of the work.

---

> ### Author Rebuttal · Authors · 2023-08-27
>
> We thank the reviewer for the useful review and the recognition of the utility of the work in making precise statements about the capabilities of language models and the limitations of simple RNNs.
>
> As pointed out, the work makes concrete limiting assumptions on the LM architecture, which, as mentioned, were discussed and justified in the limitations section. Given these assumptions, the proof is indeed not complicated. Yet, we think it is still an important contribution as it provides a novel perspective on *how* probabilities from a formal computational model can be incorporated into a more modern language model. Some of our concurrent work shows that the methodology developed in this paper can be applied to other paradigms, for example, computational models higher on the Chomsky hierarchy (e.g., pushdown automata) as well as for the analysis of Transformer models.
>
> Regarding the **question**: The “common” assumption mainly refers to classical constructions such as:
> 1. *McCulloch, W.S., Pitts, W. A logical calculus of the ideas immanent in nervous activity. Bulletin of Mathematical Biophysics 5, 115–133 (1943). https://doi.org/10.1007/BF02478259*
> 2. *Marvin Lee Minsky. 1954. Neural Nets and the brain model problem. Ph.D. thesis, Princeton University.*
> 3. *A. K. Dewdney. 1977. Threshold matrices and the state assignment problem for neural nets. In Proceedings of the 8th SouthEastern Conference on Combinatorics, Graph Theory and Computing, pages 227–245, Baton Rouge, La, USA.*
> 4. *N. Alon, A. Dewdney and T. Ott, Efficient simulation of finite automata by neural nets, Journal of Association Computing Machinery 38 (1991) 498–514.*
> 5. *P. Indyk. 1995. Optimal simulation of automata by neural nets. In STACS 95, pages 337–348, Berlin, Heidelberg. Springer Berlin Heidelberg.*
>
> The assumption on the binary nature of the activation function is also motivated by the “limiting” behavior of the standard sigmoid function, as discussed in work by William Merrill, for example in *Merrill, W. Sequential neural networks as automata.*

---

### Official Review · Reviewer_R8qp · 2023-08-05

**Soundness:** 5

**Excitement:**

4: Strong: This paper deepens the understanding of some phenomenon or lowers the barriers to an existing research direction.

**Missing References:**

* Satwik Bhattamishra, Kabir Ahuja, and Navin Goyal. 2020. On the Practical Ability of Recurrent Neural Networks to Recognize Hierarchical Languages. In Proceedings of the 28th International Conference on Computational Linguistics, pages 1481–1494, Barcelona, Spain (Online). International Committee on Computational Linguistics.
* Grégoire Delétang, Anian Ruoss, Jordi Grau-Moya, Tim Genewein, Li Kevin Wenliang, Elliot Catt, Chris Cundy et al. "Neural networks and the chomsky hierarchy." arXiv preprint arXiv:2207.02098 (2022).
* William Merrill. Formal language theory meets modern NLP. arXiv preprint arXiv:2102.10094.
* Mirac Suzgun, Yonatan Belinkov, Stuart Shieber, and Sebastian Gehrmann. 2019. LSTM Networks Can Perform Dynamic Counting. In Proceedings of the Workshop on Deep Learning and Formal Languages: Building Bridges, pages 44–54, Florence. Association for Computational Linguistics.
* Bhattamishra, Satwik, Kabir Ahuja, and Navin Goyal. "On the practical ability of recurrent neural networks to recognize hierarchical languages." arXiv preprint arXiv:2011.03965 (2020).
* Jennifer C. White and Ryan Cotterell. 2021. Examining the Inductive Bias of Neural Language Models with Artificial Languages. In Proceedings of the 59th Annual Meeting of the Association for Computational Linguistics and the 11th International Joint Conference on Natural Language Processing (Volume 1: Long Papers), pages 454–463, Online. Association for Computational Linguistics.


**Paper Topic And Main Contributions:**

This paper provides an insightful study of the expressive capabilities of simple Elman-style recurrent neural networks (Elman-RNNs) through the lens of the weighted formal language theory. It extends Minsky’s classification construction of an Elman-RNN with the Heaviside step function as its nonlinearity (also known as, Heaviside-activated Elman-RNN) simulating a finite-state automaton to a more probabilistic setup, establishing the equivalence between Heaviside-activated Elman-RNNs and deterministic finite-state automata. This result implies that Heaviside-activated Elman-RNNs can capture a rather strict subset of probability distributes expressible by deterministic finite-state automata.

Furthermore, the paper confirms a widely accepted hypothesis that a Heavisde-activated Elman-RNN capturing the same probability distribution as a deterministic probabilistic finite-state automaton with an alphabet $Σ$ and states $Q$ needs to have at least $\Omega (|\Sigma|||Q|)$ parameters (hidden states). It is important to note that this bound also corresponds to the space complexity of Minsky’s original construction.

**Recommendation**: Overall, this paper is to be commended for its insightful and clear study of the connection between Heaviside-activated Elman-style RNNs and deterministic (probabilistic) finite state automata. Considering its high-quality content, I believe that this paper fully meets the criteria for publication at EMNLP.

**Questions For The Authors:**

* Question A. Could you please clarify what you mean by the term "tight" in the context of probability distributions? (see: L163)

* Question B. (Feel free to skip this question) “For example, the very simple examples of (bounded) nesting expressible with context-free grammars are relatively awkward to express with finite-state formalisms such as finite-state automata—while they are expressible with such formalisms, the implementations lack the conciseness (and thus inductive biases) of the more concise formalisms.” (L683-690) What do you mean by “relatively awkward” in this context? Do you mean that it would be computationally expensive to express them with finite-state automata?

* Question C. Have you considered extending this framework to self-attention models?

**Reasons To Accept:**

* Overall, this paper unquestionably fulfills all the criteria for publication. It stands out not only for its impeccable writing and clarity but also for its accessible and insightful exploration of the connection between Elman-RNNs and deterministic probabilistic finite-state automata. It also includes useful asymptotic bounds on the space complexity of simulating PFSAs with Heaviside-activated Elman-RNNs.
* The formal statements, as well as the accompanying proofs, are mostly convincing and easy-to-follow.
* Moreover, the authors' thorough research is evident as they reference not only Minsky's classical construction of an RNN encoding a weighted finite-state automaton from 1954 but also works by Dewdney (1977), Indyk (1995), and Icard (2020) on RNNs, finite-state automata, and formal languages. This broadens the context and enriches the paper’s historical background, making it a delightful and well-rounded contribution to the field.


**Reasons To Reject:**

* While this is not a major ground for rejection in any way, I believe the authors could have generously acknowledged the contributions of—and insights from— more recent studies that also investigate the practical capabilities and theoretical expressivity of RNN-based architectures. Whilst this paper might indeed be the first paper to formally and explicitly study “modern [language models] with weighted formal language theory,” I believe other studies have also taken conceptually similar paths. These include:
  * William Merrill, Gail Weiss, Yoav Goldberg, Roy Schwartz, Noah A. Smith, and Eran Yahav. 2020. A Formal Hierarchy of RNN Architectures. In Proceedings of the 58th Annual Meeting of the Association for Computational Linguistics, pages 443–459, Online. Association for Computational Linguistics.
  * William Merrill. 2019. Sequential Neural Networks as Automata. In Proceedings of the Workshop on Deep Learning and Formal Languages: Building Bridges, pages 1–13, Florence. Association for Computational Linguistics.
  * Gail Weiss, Yoav Goldberg, and Eran Yahav. 2018. On the Practical Computational Power of Finite Precision RNNs for Language Recognition. In Proceedings of the 56th Annual Meeting of the Association for Computational Linguistics (Volume 2: Short Papers), pages 740–745, Melbourne, Australia. Association for Computational Linguistics.
  * Satwik Bhattamishra, Kabir Ahuja, and Navin Goyal. 2020. On the Practical Ability of Recurrent Neural Networks to Recognize Hierarchical Languages. In Proceedings of the 28th International Conference on Computational Linguistics, pages 1481–1494, Barcelona, Spain (Online). International Committee on Computational Linguistics.
  * Grégoire Delétang, Anian Ruoss, Jordi Grau-Moya, Tim Genewein, Li Kevin Wenliang, Elliot Catt, Chris Cundy et al. "Neural networks and the chomsky hierarchy." arXiv preprint arXiv:2207.02098 (2022).
  * Michael Hahn. 2020. Theoretical Limitations of Self-Attention in Neural Sequence Models. Transactions of the Association for Computational Linguistics, 8:156–171.
* I believe the paper could be improved by giving more comprehensive acknowledgments of the contributions made by other researchers in the field. While the present study itself is commendable, it would be beneficial to recognize and cite the works of relevant authors who have paved the way in areas related to Elman-RNNs, deterministic probabilistic finite-state automata, and formal languages.

**Reproducibility:**

5: Could easily reproduce the results.

**Reviewer Confidence:**

4: Quite sure. I tried to check the important points carefully. It's unlikely, though conceivable, that I missed something that should affect my ratings.

**Typos Grammar Style And Presentation Improvements:**

* Figure 1 does not feel to contribute much to the understanding of the paper. I would rather get rid of Figure 1 and promote Figure 2 to the first page.
* L261: Extra space before "ERNN"
* L265: In theory, should r be from Sigma-Bar instead of Sigma?
* L375: [D] = [1, …, D] ← might want to clarify this.

---

> ### Author Rebuttal · Authors · 2023-08-27
>
> We thank the reviewer for the thoroughly detailed and useful review as well as the nice words appreciating the writing style and the clarity of the presentation. We are also very grateful for the recognition of the thoroughness and contributions of the paper and for the recommendation of the paper for acceptance.
>
> Regarding the request for the coverage of more background material: We fully agree with the stated points. Should the paper be accepted, we will use some of the additional space for a comprehensive discussion of background material, including all the suggested work (thank you for the comprehensive list!), as well as a plethora of other works, such as *Peng et al. Rational recurrences* and *Chen et al. Recurrent neural networks as weighted language recognizers.*
> With this, we plan to cover the contributions of others working in the intersection of RNNs, finite-state automata, and other formalisms. We agree such existing work is in some ways conceptually similar and provides very valuable insights. We were initially simply hesitant to cover a large extent of the current work as it does not (apart from the work by Icard) talk about weighted formalisms.
>
> We are also grateful for the spotted typos and the raised comment about Figure 1. Indeed, we have improved Figure 2 and promoted it to the first page while also removing Figure 1.
>
> **Answers to questions**:
> 1. Regarding Question 1: “Tight”, also known as consistent, language models are autoregressive models that define a valid probability distribution over finite strings. It is relatively simple to construct PFSAs that put probability mass on infinite sequences, resulting in a model where the probability mass on all *finite* strings is $<1$ instead of $1$. See *Du et al. A Measure-Theoretic Characterization of Tight Language Models* for a comprehensive study and discussion of the tightness of language modes.
> 2. Regarding Question 2: Your interpretation is correct. For example, the Dyck(k) language admits a very simple description with a context-free grammar, while the description of the Dyck(k, m) language using an FSA requires a large automaton covering every “case” separately in a separate state.
> 3. Regarding Question 3: Studying self-attention is the natural next step, yes. We have done some preliminary investigation into this and while we believe similar results can be achieved, the constructions are more elaborate and nuanced; due to the parallelizability, self-attention does not keep an internal state, which means that the connection to sequential models such as FSAs is more difficult. We, however, believe it can be done by augmenting the output alphabet and plan to showcase our constructions in future work.

---

### Meta-Review · Area_Chair_sVvB · 2023-09-12

**Recommendation:** 5

**Metareview:**

The reviewers are in broad consensus that this paper makes a substantial contribution to our understanding of the capabilities of RNNs. They agree that the insight is valuable, the investigation is thorough, and the paper is well written. While Reviewer uUpK points to some simplifying assumptions that may be considered a bit strong, but the authors openly address these limitations in the paper. The extensive discussion in the appendix might make this work better suited as a journal paper, but if the authors prefer to keep that discussion in the appendix and publish in a conference, I see no good reason to stop them.

I hope the authors incorporate the presentation and literature review suggestions in their paper revisions.

---

### Decision · Program_Chairs · 2023-10-07

**Decision:**

Accept-Main

**Comment:**

The reviewers are in broad consensus that this paper makes a substantial contribution to our understanding of the capabilities of RNNs. They agree that the insight is valuable, the investigation is thorough, and the paper is well written. While Reviewer uUpK points to some simplifying assumptions that may be considered a bit strong, but the authors openly address these limitations in the paper. The extensive discussion in the appendix might make this work better suited as a journal paper, but if the authors prefer to keep that discussion in the appendix and publish in a conference, I see no good reason to stop them.

I hope the authors incorporate the presentation and literature review suggestions in their paper revisions.